# Duration of Antibiotic Treatment for Foot Osteomyelitis in People with Diabetes

**DOI:** 10.3390/antibiotics13121173

**Published:** 2024-12-04

**Authors:** Meryl Cinzía Tila Tamara Gramberg, Bart Torensma, Suzanne van Asten, Elske Sieswerda, Louise Willy Elizabeth Sabelis, Martin den Heijer, Ralph de Vries, Vincent de Groot, Edgar Josephus Gerardus Peters

**Affiliations:** 1Division of Infectious Diseases, Department Internal Medicine, Amsterdam UMC, Vrije Universiteit Amsterdam, 1081 HV Amsterdam, The Netherlands; 2Department of Rehabilitation Medicine, Amsterdam UMC, Vrije Universiteit Amsterdam, De Boelelaan 1117, 1081 HV Amsterdam, The Netherlands; 3Amsterdam Movement Sciences, Rehabilitation and Development, 1081 HV Amsterdam, The Netherlands; 4Amsterdam Infection & Immunity, Infectious Diseases, 1081 HV Amsterdam, The Netherlands; 5Amsterdam UMC Center for Diabetic Foot Complications (ACDC), 1081 HV Amsterdam, The Netherlands; 6Clinical Epidemiologist, Leiden University Medical Centre, 2333 ZA Leiden, The Netherlands; 7Department of Medical Microbiology, Medical Microbiology, Radboud UMC, 6525 GA Nijmegen, The Netherlands; 8Julius Center for Health Sciences and Primary Care, UMC Utrecht, 3584 CX Utrecht, The Netherlands; 9Department of Medical Microbiology, UMC Utrecht, 3584 CX Utrecht, The Netherlands; 10Division of Endocrinology, Department of Internal Medicine, Academisch Medisch Centrum, Amsterdam UMC, Vrije Universiteit Amsterdam, 1081 HV Amsterdam, The Netherlands; 11University Library, Vrije Universiteit Amsterdam, 1081 HV Amsterdam, The Netherlands; r2.de.vries@vu.nl

**Keywords:** diabetes-related foot osteomyelitis, antimicrobial therapy, duration, short versus long

## Abstract

**Background:** The optimal antimicrobial treatment duration for diabetes-related foot osteomyelitis (DFO) currently needs to be determined. We systematically reviewed the effects of short and long treatment durations on outcomes of DFO. **Methods:** We performed a systematic review searching Cochrane, CENTRAL, MEDLINE, Embase, and CINAHL Plus from inception up to 19 January 2024. Two independent reviewers screened the titles and abstracts of the studies. Studies comparing short (<6 weeks) and long (>6 weeks) treatment durations for DFO were included. The primary outcome was amputation; the secondary outcomes were remission, mortality, costs, quality of life, and adverse events. Risk of bias and GRADE were assessed. **Results:** We identified 2708 references, of which 2173 remained after removing duplicates. Two studies were included. Differences in methodology precluded a meta-analysis. The primary outcome, major amputation, was reported in one study, with a rate of 10% in both the intervention and comparison groups (*p* = 1.00), regardless of treatment duration. For the secondary outcome, remission rates, the first study reported 60% in the intervention group versus 70% in the comparison group (*p* = 0.50). In the second study, remission rates were 84% in the intervention group versus 78% in the comparison group (*p* = 0.55). Data for the outcomes mortality, costs, and quality of life were not available. Short treatment duration may lead to fewer adverse events. The risk of bias was assessed as low to moderate, and the level of evidence ranged from very low to moderate. **Conclusions:** Our findings suggest that for DFO, there is no difference between a shorter and more prolonged duration of antimicrobial treatment regarding amputation and remission, with potentially fewer adverse events with shorter treatment durations. However, the uncertainty stems from limited, heterogeneous studies and generally low-quality evidence marred by moderate biases, imprecision, and indirectness. More high-quality studies are needed to substantiate these findings.

## 1. Introduction

Diabetes-related foot osteomyelitis (DFO) is a frequent and severe complication in patients with diabetes [1]. Current estimates indicate that around 131 million people (1.8% of the global population) have diabetes-related lower extremity complications such as neuropathy, peripheral artery disease, ulceration and/or infection of the foot [2]. Twenty percent of people with foot complications also have osteomyelitis, which is a leading cause of adverse clinical outcomes such as lower extremity amputation and loss of quality of life [3]. Approximately 50% to 77% of people with DFO ultimately need amputation of the lower extremity despite targeted and long-term antimicrobial therapy [4,5,6].

In short, the pathogenesis of DFO involves structural alterations and foot deformities due to persistent hyperglycemia. A combination of neuropathy (including loss of foot sensation), immune dysfunction, peripheral artery disease, and foot deformities predisposes individuals to ulcer formation and subsequent ulcer infection. If left untreated, the infection can eventually spread to the bone [7,8].

The cornerstone of conservative treatment for DFO includes systemic antimicrobial therapy, often combined with surgical intervention (e.g., ulcer debridement and/or partial amputation). In addition, optimal management of diabetes, wound care, and offloading of pressure from the affected foot are crucial components of treatment. Systemic antimicrobial therapy is preferably based on accurate cultures obtained before initiating treatment [1,9,10]. However, the effectiveness of this therapy can be compromised by difficulties in delivering antibiotics to areas with necrosis, poor blood flow, and the formation of biofilms on bone sequesters, which protect bacteria from both antibiotic action and immune responses [11].

Due to the pathophysiology of diabetes-related foot osteomyelitis (DFO), this infection generally requires surgery and systemic antibacterial therapy [1]. However, the optimal duration for systemic antibacterial therapy (sATD) remains unclear [12,13]. The goal of treatment is to address the signs and symptoms of infection, eradicate bacterial pathogens in the foot, and prevent further tissue destruction. Culture-guided treatment with an appropriate duration can lower the risk of adverse outcomes such as hospitalization, sepsis, amputation, or death [1,14].

International treatment guidelines generally suggest a treatment duration of approximately six weeks. Longer durations may be recommended for patients who do not undergo surgical debridement of the infected bone, while shorter durations are suggested for those who receive complete surgical debridement or amputation of the infected bone [1]. The guidelines emphasize tailoring the duration of therapy based on the extent of surgical intervention and the patient’s clinical response.

Shorter courses of antibiotics may offer reduced adverse effects (e.g., hepatotoxicity, acute kidney injury, skin rash, or *Clostridioides difficile*-associated diarrhea) and improved patient adherence compared with longer treatment durations [15]. This approach aligns with findings from studies on other types of infections, where shorter durations are as effective as traditional longer courses [16,17]. However, studies on other types of infections indicate that there is a potential for increased risk of infection recurrence and other negative outcomes, such as prolonged hospitalization, increased healthcare costs, development of antibiotic resistance, and higher morbidity and mortality rates, if the infection is not fully resolved [18]. Therefore, further research is needed to identify the most effective and safest sATD for DFO, considering both the potential for improved patient outcomes and the minimization of treatment-related adverse effects. In this systematic review (SR), the objective was to compare the effects of a shorter antimicrobial treatment duration with a longer one in people with diabetes-related foot osteomyelitis, aiming to assess the efficacy and safety of systemic antibacterial therapy.

## 2. Results

### 2.1. Literature Search

The literature search generated 2708 references: 1231 in Ovid Medline, 818 in Embase.com, 422 in the Cochrane Library, 148 in CINAHL, 26 in CT.gov, and 63 in ICTRP. After removing duplicates, 2173 unique references remained. After title and abstract selection, and full read, two articles were included (PRISMA flow, Figure 1).

### 2.2. Characteristics of the Included Studies

The RCTs included in this review were published in 2015 and 2021, respectively, and involved 133 participants Tone 2015 [19] and Gariani 2021 [20].

Follow-up duration varied, with Tone (2015) [19] providing 12 months of follow-up and Gariani (2021) [20] providing two months after treatment for primary outcomes. Tone (2015) [19] included 40 participants, evenly split between the intervention and comparison groups. In contrast, Gariani (2021) [20] included 93 participants, with a different distribution for intention-to-treat and per-protocol analyses.

### 2.3. Interventions and Comparisons

Table 1 provides a comprehensive overview of the participant characteristics and study details, highlighting differences in culture types, treatment durations, and the use of partial amputation. Antimicrobial treatment was based on bone culture in one study Tone (2015) [19] and tissue or bone culture in another Gariani (2021) [20]. The treatment duration was six versus 12 weeks in Tone (2015) [19] and 3 versus 6 weeks in Gariani (2021) [20]. Although both studies included ulcer debridement as standard care, partial amputation was permitted only in Gariani (2021) [20]. Neither study reported on mortality, quality of life, or treatment costs. Due to the essential differences mentioned above, there was clinically important heterogeneity. Therefore, we performed an evidence synthesis without meta-analyses.

### 2.4. Primary Outcome, Secondary Outcomes, Remission, Surgical Interventions, and Adverse Events

One study reported on the primary outcome of amputation, with a rate of 10% in both the intervention and comparison groups (*p* = 1.00). Both studies reported remission rates, which ranged from 60% to 84%, with no significant difference between shorter and longer treatment durations. Adverse events were less common in the intervention group, with significant differences observed in one study (*p* = 0.04) (Table 2 for a complete description of the results).

### 2.5. Risk of Bias Assessment

Both studies showed a low risk of bias regarding randomization and clear outcome definitions, but deviations from intended interventions in Gariani (2021), including participant exclusions and late amputations, raised concerns. Additionally, the short and inconsistent follow-up in Gariani’s study may have led to misclassification of remission and overlooked late clinical failures. These factors contribute to imprecision in the treatment effects. Overall, the risk of bias was low to moderate, with the main issues arising from deviations in intervention and variability in outcome measurement. These biases suggest that while the evidence offers valuable insights, caution is needed when interpreting the results, as these factors may impact the reliability and generalizability of the findings (Appendix C).

### 2.6. GRADE

We used the GRADE methodology to systematically assess the quality of the evidence. We rated the evidence as low, moderate, or high. The quality of evidence was downgraded, i.e., because of a risk of bias, inconsistency, indirectness, imprecision, or publication bias.

## 3. Discussion

This systematic review, despite a comprehensive and thorough search strategy, identified only two randomized controlled trials (RCTs) assessing the efficacy of shorter versus longer antimicrobial treatment durations for DFO. The limited number of studies highlights this area’s scarcity of high-quality evidence.

Due to substantial heterogeneity between the studies, a meta-analysis was not feasible. Our findings suggest that, for DFO, shorter antibiotic treatment (6 weeks) may be as effective as longer treatment (12 weeks) to achieve remission. However, the primary outcome of amputation showed no difference between groups, with a 10% rate in both the intervention and comparison arms of one study. Adverse events were less frequent in the shorter treatment group, with a significant difference noted in one study. The findings should be interpreted with caution when considering the risk of bias.

Strengths of the studies include clear outcome definitions and comprehensive follow-up, but limitations such as deviations from intended interventions, short follow-up periods, and lack of confidence intervals reduce the reliability of the results. These biases suggest that while shorter treatment may be promising, the evidence remains moderate, and further high-quality trials are needed to confirm these findings.

Furthermore, the studies’ geographical distribution, spanning Europe (France and Switzerland), suggests applicability primarily to Western countries with advanced healthcare systems; however, generalization to regions with differing healthcare qualities is cautioned.

Quality of evidence ranged from moderate for the outcome of amputation to very low for remission, influenced by concerns over bias, indirectness, and imprecision among the included studies. Our review adhered to Cochrane protocols to minimize potential biases in the review process; however, ongoing and unpublished trials represent a potential source of bias that could affect our conclusions.

Comparative literature suggests that shorter antibiotic courses may be safe for other forms of osteomyelitis and infections, such as vertebral and pediatric osteomyelitis, and infections involving prosthetic joints and soft tissues [16,17,21]. However, given the unique challenges faced by individuals with diabetes, such as peripheral artery disease and immune dysfunction, caution is needed before extrapolating these findings to the DFO population. This underscores the need for more high-quality, targeted studies. Senneville et al. may help address this evidence gap and provide clearer guidance on the safety and efficacy of shorter antibiotic regimens for DFO [1]. In the context of our findings, the limited number of high-quality RCTs makes it difficult to draw definitive conclusions regarding the optimal treatment duration for DFO. While our review indicates that shorter treatment may be as effective as longer treatment, the small sample size and moderate risk of bias in the available studies limit the strength of this conclusion. More robust evidence is needed to determine whether these findings can be generalized to the broader DFO population, especially in light of the clinical complexity of these patients.

A multidisciplinary approach that includes not only antimicrobial therapy but also standardized protocols for offloading, vascular optimization, wound care, and glycemic control is essential in the management of DFO. These treatment components can significantly influence outcomes, and future studies on antibiotic treatment duration should ensure that these elements are consistently applied and standardized across both intervention and control groups.

### 3.1. Recommendations for Future Research

To advance the understanding of optimal treatment duration for DFO, future studies should adopt standardized treatment protocols that include antimicrobial therapy and key components such as offloading, vascular optimization, wound care, and glycemic control. Ensuring that these elements are uniformly applied in both intervention and control groups is crucial for accurately assessing the effect of treatment duration on outcomes. In addition to clinical outcomes, future research should incorporate a comprehensive set of outcome measures, including patient-reported outcomes and quality-of-life assessments, vital for patient-centered care. Primary outcomes should include remission rates and amputation necessity, while secondary outcomes could examine antibiotic resistance, adverse events, and long-term functional outcomes. Economic evaluations, such as cost-effectiveness analyses, should also be integrated to assess the feasibility of different treatment durations in various healthcare settings, particularly in resource-limited environments. Additionally, future research should incorporate subgroup analyses to explore how patient-specific factors, such as age, gender, diabetes type, osteomyelitis severity, and peripheral arterial disease, may impact the optimal treatment duration. By ensuring that these key factors are uniformly managed and documented, future studies can provide more accurate and tailored guidelines for the management of DFO, leading to more effective and individualized treatment strategies. Furthermore, the impact of antimicrobial duration on microbial resistance patterns warrants discussion. With the global rise in antibiotic resistance, understanding how shorter treatment regimens might mitigate this risk is crucial for guiding future treatment protocols. Finally, studies should ensure adequate follow-up periods to capture late clinical failures and recurrences. Subgroup analyses, considering factors like age, gender, diabetes type, and osteomyelitis severity, can help tailor treatment guidelines to diverse patient populations. By addressing these gaps, future research can provide more precise and globally applicable recommendations for managing DFO. These limitations underscore the need for further research with larger, more diverse populations and standardized outcome measures. Future studies should ensure that treatment protocols include antibiotic therapy and essential components, all applied consistently across intervention and control groups. Additionally, incorporating long-term follow-up is critical to assess the sustainability of treatment effects, particularly regarding recurrence rates, antibiotic resistance, and long-term functional outcomes. Future research should equally include patient-reported outcomes and quality-of-life measures, as these provide valuable insights into the patient experience that clinical outcomes alone cannot capture. Economic evaluations, such as cost-effectiveness analyses, are integral in assessing the feasibility of different treatment durations for DFO across diverse healthcare settings. By understanding the potential cost savings associated with shorter antibiotic courses—while maintaining clinical outcomes—healthcare providers and policymakers can make informed decisions that balance effective patient care with resource allocation. This approach may guide policy recommendations for antimicrobial stewardship and optimize treatment strategies, particularly in resource-constrained settings where prolonged antibiotic use may present additional financial and logistical burdens. Subgroup analyses could help tailor treatment guidelines for specific patient populations. Addressing these gaps will contribute to a more comprehensive and globally applicable understanding of the optimal duration of antibiotic treatment for diabetic foot osteomyelitis, improving patient care and treatment outcomes in diverse clinical settings.

### 3.2. Limitations

Addressing the limitations of our systematic review is crucial for understanding the context and scope of our findings. A significant limitation stems from the small sample sizes of the included studies, which precluded the possibility of conducting a meta-analysis. This constraint limits the statistical power and generalizability of our conclusions, as the heterogeneity and variability inherent in the study designs, populations, and interventions could not be quantified or systematically compared. Another critical limitation is the challenge of adequately controlling for confounding factors. Given the observational nature of some of the included data and the variability in study designs, fully adjusting for all potential confounders that could influence the outcomes of antibiotic duration for diabetes-related foot osteomyelitis was not feasible. This could lead to residual confounding, affecting the validity of the findings. Additionally, the heterogeneity in outcome definitions and measurement across studies introduces variability that complicates the synthesis of results. This inconsistency makes it challenging to draw firm conclusions about the efficacy and safety of shorter versus longer antibiotic treatments. Furthermore, the lack of long-term follow-up in the included studies restricts our understanding of the sustainability of treatment effects, particularly concerning recurrence rates, resistance development, and long-term patient outcomes. The absence of patient-reported outcomes in the reviewed literature is another limitation, as it overlooks the subjective experience of the disease and treatment, which is crucial for a holistic understanding of treatment impact.

And lastly, the current antimicrobial treatments for DFO, which extend beyond treatment duration, are the growing concern of antibiotic resistance. Prolonged or repeated antibiotic courses can contribute to resistance, limiting future treatment options and potentially decreasing the efficacy of standard regimens. Moreover, the presence of biofilms—structured bacterial communities that adhere to tissue and implant surfaces—presents a significant barrier to successful treatment. Biofilms protect bacteria from immune responses and limit antibiotic penetration, complicating eradication and often necessitating more aggressive or extended therapies. These challenges underscore the need for more tailored treatment strategies to address antibiotic resistance and biofilm-associated infections, which are frequent and problematic in DFO patients. Future research should explore alternative or adjunct therapies to overcome these limitations and improve treatment outcomes.

## 4. Materials and Methods

We conducted a SR in accordance with the PRISMA (Reporting Items for Systematic Reviews and Meta-Analyses guidelines) by Moher et al. [22] (Checklist Appendix A). The objective of this SR was to compare the effects of a shorter antimicrobial treatment duration with a longer antimicrobial treatment duration in people with diabetes-related foot osteomyelitis.

### 4.1. Search Strategy

To identify the relevant publications, we conducted systematic searches in the bibliographic databases Ovid Medline, Embase.com, Wiley/Cochrane Library, and Ebsco/CINAHL from inception to 19 January 2024, in collaboration with a medical information specialist. We used the following terms (including synonyms and closely related words) as index terms or free-text words: “Osteomyelitis”, “Osteitis,” “Diabetic foot”, “Anti-Bacterial Agents”, and “randomized controlled trials”. The full search strategies for all databases can be found in Appendix B. Additionally, we searched ClinicalTrials.gov and ICTRP. Also, we searched grey literature and performed a reference crosscheck to identify eligible articles not identified through previous searches. There were no restrictions on language or publication date. We excluded duplicate articles using Endnote X21.0.1 (Clarivatetm).

### 4.2. Study Eligibility Criteria

Only randomized controlled trials (RCTs) were included. Cohort, cross-sectional studies, case–control studies, and case reports were excluded.

### 4.3. Outcomes

The primary outcome was amputation, including part of the foot (i.e., minor, distal to the malleoli), leg (i.e., major, proximal to the malleoli), or complete bone resection. The secondary outcomes are clinical remission; surgical intervention of the foot, including surgical debridement and partial bone resection without removing the complete bone or performing an amputation; adverse events, expressed as the proportion of participants in each group with an event; mortality; and costs.

### 4.4. Types of Participants

Participants included patients with type 1 or 2 diabetes mellitus (DM) and osteomyelitis distal to the malleoli.

### 4.5. Study Selection and Data Extraction

Two independent reviewers (MG, SvA) screened the studies’ titles and abstracts using the inclusion and exclusion criteria in the Covidence systematic review software (Veritas Health Innovation, Melbourne, Australia). Subsequently, the same reviewers independently reviewed the remaining full-text reports for eligibility. Data from the full-text articles were extracted independently. In all stages, we resolved disagreements by discussing or consulting a third independent reviewer (EP).

### 4.6. Assessment of Risk of Bias

Two reviewers (MG, BT) independently assessed the risk of bias (RoB) for the methodological quality of each included study using the PRISMA guidelines Moher et al. [22] and the Cochrane risk of bias tool for randomized trials (RoB 2). We tested the RoB on the domains of randomization, deviations from intended interventions, missing outcome data, measurement of outcome, and selection of the reported results. We scored all domains from low to high risk of bias.

### 4.7. Statistical Analysis

We extracted all studies, and we displayed the characteristics of the variables displayed to the definitions used in the articles. Heterogeneity was evaluated using the tau-squared (I2) statistic. I2 of 0–40% was considered as low heterogeneity, 30–50% as moderate heterogeneity, 50–75% as substantial heterogeneity, and 75–100% as high heterogeneity, respectively. When heterogeneity was greater than 60% or if definitions of the outcome, or outcomes were not displayed, a meta-analysis was not performed. We set statistical significance at *p* < 0.05 and performed all analyses using Review Manager version 5.4.1 (Review Manager 2020 [Computer program]).

## 5. Conclusions

The limited number of studies and participants, combined with substantial risk of bias and the absence of data on key outcomes such as mortality, treatment costs, and quality of life, restricts the scope of this review and the completeness and applicability of the available evidence. Our findings suggest that a shorter duration of antimicrobial treatment for DFO may be as effective as a longer one in terms of amputation and remission, with fewer adverse events. However, the limited number of studies, their heterogeneity, and overall low quality of evidence—characterized by biases, imprecision, and indirectness—necessitate caution in drawing firm conclusions. More high-quality research is needed to define the optimal treatment duration better.

## Figures and Tables

**Figure 1 antibiotics-13-01173-f001:**
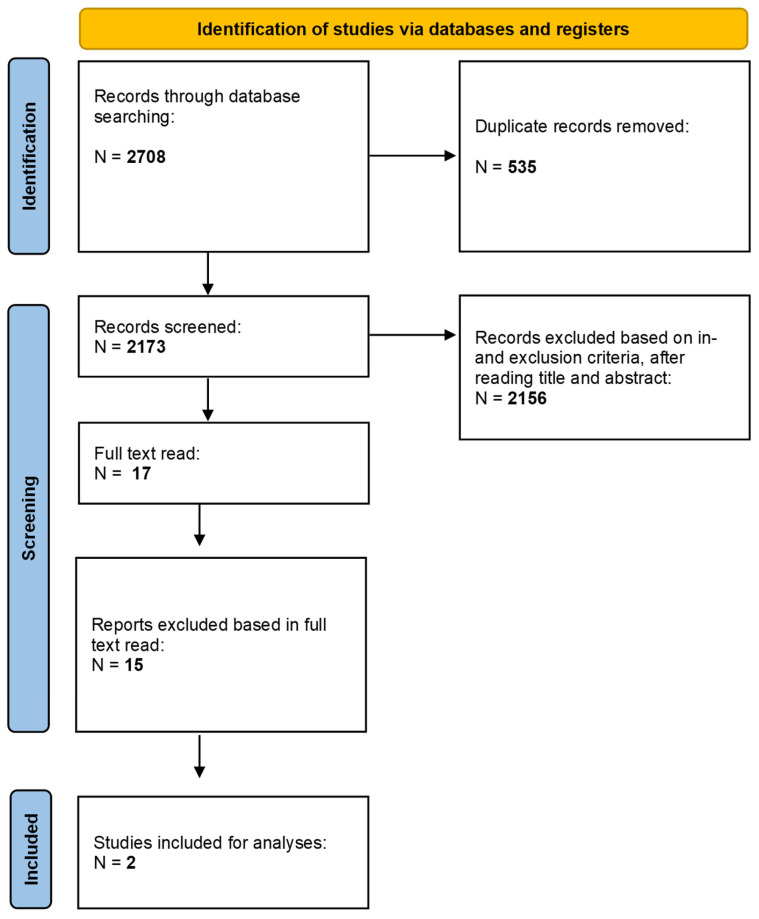
PRISMA flow chart.

**Table 1 antibiotics-13-01173-t001:** Study characteristics.

Author	Tone 2015 [19]	Gariani 2021 [20]
Study design	Randomized controlled trial, parallel group	Randomized controlled trial, parallel group
Duration of study	3 years	3 years
Outcome assessment	12 months	2 months after end of treatment
Interventions (Group 1)	Antimicrobial treatment: 6 weeks Clinician could select the most appropriate	Antimicrobial treatment: 3 weeks Clinician could select the most appropriate
Comparison (Group 2)	Antimicrobial treatment: 12 weeks	Antimicrobial treatment: 6 weeks
Number of participants	*n* = 40 Group 1 *n* = 20 Group 2 *n* = 20	*n* = 93 Group 1 *n* = 44 Group 2 *n* = 49
Age (years)	Group 1: 64.6 ± 11.2 Group 2: 63.8 ± 10.9	Group 1 *n* = 70 ± not mentioned Group 2 *n* = 65 ± not mentioned
Sex ratio (male/female) *n*=	11/29	76/17
Type of diabetes	Type 2, duration > 10 years in 73% of participants	Type 1 and 2 diabetes mellitus Not specified in percentages
Pre-disease duration	Not specified	Not specified
Inclusion criteria	Included: ≥18 years DFO was suspected with two of the following: wound > 2 weeks over bony prominence, ulcer surface > 2 cm^2^ or depth > 3 mm, probing to bone, or imaging abnormalities consistent with osteomyelitis. Confirmation required a positive culture from a transcutaneous bone biopsy after a 2-week antibiotic-free period.	Included: ≥18 years, DFO DFO per Infectious Diseases Society of America DFI guidelines and including suggestive radiological, clinical, and/or bone microbiological findings
Exclusion criteria	Absence of both pedal pulses, gangrene, severe peri-osteoarticular damage	DFO associated with an implant; recent antibiotic therapy within 96 h; total amputation of infected bone; complete bone destruction beyond cortical level; or remote infection needing over 21 days of another antibiotic therapy.
Outcome	Remission (1) No signs of infection, (2) stable/improved X-ray abnormalities at end of treatment and 1 year later, and (3) complete wound healing. Remission required no relapse or surgery at the initial site or contiguous rays for at least 12 months.	Remission Complete absence of clinical and radiologic findings of infection after a minimum follow-up of 2 months after end of treatment.
Outcomes: mortality, costs, and quality of life	No information	No information

DFO = diabetes-related foot osteomyelitis. AEs = adverse events.

**Table 2 antibiotics-13-01173-t002:** Study outcomes and complications.

Author	Tone 2015 [19]	Gariani 2021 [20]
Remission	Group 1 * 12 (60%) Group 2 ** 14 (70%) (*p* = 0.50)	Group 1 * 37 (84%) Group 2 ** 36 (73%) (*p* = 0.21)
Surgical interventions	No information. The study did not permit partial amputation within its protocol.	The median number of surgical debridement procedures per episode was 1 (IQR, 0–2), of which 34 (36%) involved partial amputations.
Major amputation	Group 1: 2 (10%) Group 2: 2 (10%)	No information
Overall failure	Group 1: 8 (40%) Group 2: 6 (30%)	Group 1: 3 (3.2%) Group 2: 5 (5.4%)
Noncomplete healing	Group 1: 2 (10%) Group 2: 4 (20%)	No information
Relapsing osteomyelitis	Group 1: 2 (10%) Group 2: 3 (15%)	No information
Worsening radiological bone abnormalities	Group 1: 6 (30%) Group 2: 4 (20%)	No information
Bone resection	Group 1: 2 (10%) Group 2: 2 (10%)	No information
Spread of osteomyelitis to contiguous sites	Group 1: 4 (20%) Group 2: 2 (10%)	No information
Nausea	Group 1: 1 (5%) Group 2: 2 (10%)	1 (1.1%)—treatment group not specified
Vomiting	Group 1: 1 (5%) Group 2: 2 (10%)	No information
Diarrhoea	Group 1: 0 (0%) Group 2: 2 (10%)	1 (1.1%)—treatment group not specified
Fungal intertrigo	No information	4 (4.3%)—treatment group not specified
Anaphylaxis	No information	1 (1.1%)—treatment group not specified
Drug fever	No information	1 (1.1%)—treatment group not specified
Skin rash	No information	3 (3.2%)—treatment group not specified
Hepatic cytolysis/cholestasis	Group 1: 1 (5%) Group 2: 3 (15%)	No information

* Group 1 is the group with a shorter treatment duration. ** Group 2 is the group with a longer treatment duration.

## Data Availability

No new data were created or analyzed in this study. Data sharing is not applicable to this article.

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
