# Peer review of "Duration of Antibiotic Treatment for Foot Osteomyelitis in People with Diabetes"

_antibiotics, 2024, doi:10.3390/antibiotics13121173_

Round 1
Reviewer 1 Report
Comments and Suggestions for Authors
Title: Duration of antibiotic treatment for foot osteomyelitis in people with
diabetes
1. Figure 1 is missing in the manuscript.
2. As only two studies were included, the conclusions of the review may be limited in scope and generalizability. Conducting a systematic review with only two studies presents a significant challenge, as it may result in bias and a lack of comprehensive evidence, especially on a topic like antibiotic treatment for foot osteomyelitis in people with diabetes. Systematic reviews are designed to offer a comprehensive synthesis of the literature, which can prove challenging when dealing with a limited number of two studies for analysis and interpretation.
Reviewer 2 Report
Comments and Suggestions for Authors
The authors aim to conduct a systematic review to analyze the efficacy and safety of the duration of systemic antibacterial therapy. The review adheres to the principles of the PRISMA methodology, ensuring a transparent evaluation of the available evidence. The introduction provides a clear and concise overview of diabetic foot osteomyelitis (DFO), highlighting its prevalence, severity, and impact on patients' quality of life. Additionally, the discussion presents a solid and critical analysis of the findings, acknowledging the study's limitations and suggesting future directions for research.
Based on the above, it is recommended to accept this work, provided that the following minor comments are addressed:
- Although the introduction adequately establishes the context and importance of DFO, the authors should clarify their objective. It would be beneficial to explicitly state that the purpose of the work is to compare both the efficacy and safety of the duration of systemic antibacterial therapy. Moreover, the limitations of current treatments, such as antibiotic resistance and the challenges in eradicating biofilms, could be highlighted.
- It is important to ensure that all abbreviations are defined, for example, the abbreviation "sATD."
- A PRISMA Flow Diagram is mentioned in Figure 1, but it does not appear. It is necessary to include it.
- Tables should not only be numbered but also have a title that adequately describes them.
- The phrase "from the beginning" is ambiguous; the exact date when the search commenced should be specified.
- The title would benefit from specifying that it is a systematic review.
- The results and the analysis of cost and effectiveness are well integrated; however, it may be worth discussing their significance in clinical decision-making and health sector policies.
- While the use of gray literature is mentioned, the authors should indicate the sources consulted for this literature.
Reviewer 3 Report
Comments and Suggestions for Authors
The study reviews antimicrobial treatment durations for diabetes-related foot osteomyelitis (DFO), comparing short (<6 weeks) and long (>6 weeks) durations. Here are some of my observations:
- The search process used to perform literature search is detailed. The supplementary material is also helpful.
- The reason why I am a bit concerned is because the systematic analysis only deals with two studies. Both these studies are small in size and one of these studies is close to a decade ago.
- Very few outcomes were observed in both studies. It is difficult to make a comparison between the two studies.
- I agree that the goal was to do a meta-analysis but due to the small (just 2) the authors settled for a systematic analysis.
- I assume that both of these studies were performed in developed countries. The burden of diabetes is larger in developing countries. Have the authors explored that?
Round 2
Reviewer 1 Report
Comments and Suggestions for Authors
The authors have thoroughly revised the manuscript and addressed the questions I raised during the review process. Their responses demonstrate a clear understanding of the issues and enhance the overall quality of the work. I recommend accepting the manuscript for publication, as it contributes significantly to the field.
Author Response
Thank you for your time and commitment reviewing our manuscript.
Reviewer 2 Report
Comments and Suggestions for Authors
The authors aim to conduct a systematic review to assess the efficacy and safety of varying durations of systemic antibacterial therapy. This review adheres to PRISMA methodology principles, ensuring a transparent evaluation of the available evidence. The introduction provides a clear and concise overview of diabetic foot osteomyelitis (DFO), highlighting its prevalence, severity, and impact on patients' quality of life. Furthermore, the discussion offers a thorough and critical analysis of the findings, acknowledging the study’s limitations and suggesting future research directions. All suggested revisions have been incorporated. Therefore, it is recommended that the article be accepted for publication.
Author Response

(The authors gave the same response as above.)

Reviewer 3 Report
Comments and Suggestions for Authors
Thanks for giving me a chance to review. The authors need to be honest about the fact that only two studies (that one too more than a decade old) were analyzed when writing this article. This limits the quality and generalizability of the study.
Author Response
Dear reviewer,
Thank you for your time and committment reviewing our manuscript. We underline this limitation of the study in our discussion. In our discussion and limitations section we acknowledge this shortcoming, but when performing a systematic search we found that this is the current body of evidence.